# Prevalence of Microalbuminuria and Its Association with Subclinical Carotid Atherosclerosis in Middle Aged, Nondiabetic, Low to Moderate Cardiovascular Risk Individuals with or without Hypertension

**DOI:** 10.3390/diagnostics11091716

**Published:** 2021-09-19

**Authors:** Eva Szabóová, Alexandra Lisovszki, Eliška Fatľová, Peter Kolarčik, Peter Szabó, Tomáš Molnár

**Affiliations:** 1Department of Angiology, Faculty of Medicine, East Slovak Institute of Cardiovascular Diseases, Pavol Jozef Šafárik University, 040 01 Košice, Slovakia; 24th Department of Internal Medicine, Faculty of Medicine, Louis Pasteur University Hospital, Pavol Jozef Šafárik University, 040 01 Košice, Slovakia; alex.kocsis@centrum.sk (A.L.); fatlova.eliska@yahoo.com (E.F.); 3Department of Health Psychology and Research Methodology, Faculty of Medicine, Pavol Jozef Šafárik University, 040 01 Košice, Slovakia; peter.kolarcik@upjs.sk; 4Department of Aviation Technical Studies, Technical University of Košice, 040 01 Košice, Slovakia; peter.szabo@tuke.sk; 5Department of Vascular Surgery, Faculty of Medicine, East Slovak Institute of Cardiovascular Diseases, Pavol Jozef Šafárik University, 040 01 Košice, Slovakia; tomas.molnar@vusch.sk

**Keywords:** microalbuminuria, subclinical atherosclerosis, carotid ultrasound, carotid intima–media thickness, carotid plaque, arterial stiffness

## Abstract

Microalbuminuria is closely associated with the risk of cardiovascular disease and all-cause mortality in the general population. Less is known about its relationship with subclinical atherosclerosis. We aimed to assess the prevalence of microalbuminuria and its relationship with subclinical atherosclerosis in middle-aged, nondiabetic, apparently healthy individuals (N = 187; 40.1% men, 59.9% women; aged 35–55 years) as well as to evaluate its potential associations with established risk modifiers, especially with the presence of carotid plaque. Clinical and laboratory parameters, the estimated 10-year fatal cardiovascular risk (SCORE), as well as circulating, functional (flow mediated vasodilation, ankle-brachial index, augmentation index, and pulse wave velocity), and morphological markers (mean carotid intima–media thickness, and carotid plaque) of subclinical atherosclerosis were analysed in group with vs. without microalbuminuria. Microalbuminuria was present in 3.8% of individuals with SCORE risk 0.43 ± 0.79%. Functional markers predominated in both groups. Carotid intima–media thickness (mean ± SD) in both groups was in range: 0.5–0.55 ± 0.09–0.14 mm. Carotid plaque was more frequent in group with (14.3%) vs. without (4.4%) microalbuminuria. Microalbuminuria had no statistically significant effect on most markers of subclinical atherosclerosis, but the increasing value of microalbuminuria was significantly associated with the occurrence of carotid plaque (*p* = 0.035; OR = 1.035; 95% CI = 1.002–1.07). Additional multiple logistic regression analysis, where variables belonged to microalbuminuria, number of risk factors, and family history, finally showed only two variables: microalbuminuria (*p* = 0.034; OR = 1.04; 95%CI = 1.003–1.09) and the number of risk factors (*p* = 0.006; OR = 2.15; 95% CI = 1.24–3.73) with independent and significant impact on the occurrence of carotid plaque. Our results may indicate an association of microalbuminuria with the presence of carotid atherosclerotic plaque; in addition, microalbuminuria and the number of risk factors appear to be possible predictors of the carotid plaque occurrence. Monitoring microalbuminuria may improve the personalized cardiovascular risk assessment in nondiabetic, low-to-moderate cardiovascular risk individuals with or without hypertension.

## 1. Introduction

Microalbuminuria is closely associated with renal and cardiovascular (CV) morbidity and mortality in diabetic [1], hypertensive [2], and elderly patients [3], but predicts also all-cause mortality in the general population [4]. Moreover, microalbuminuria is associated and clustered with widely known CV risk factors [5]. The predictive power of urinary albumin levels for CV risk is independent of other CV risk factors and is present also in healthy individuals [5]. Albuminuria is used as an indicator of generalized endothelial dysfunction, an early stage of atherosclerosis [5]. To detect preclinical atherosclerosis, biochemical (blood and urinary, including albuminuria), functional, and morphological markers are used. Some widely available hematological and coagulation parameters can help in risk stratification of patients with acute coronary syndromes, some red blood cell and platelet parameters may indicate proatherogenic lipoprotein profiles in the general population [6,7,8]. Endothelial dysfunction can be measured by endothelium-dependent, flow mediated vasodilation (FMD), which is currently considered to be the gold standard functional test, but has high measurement variability [9]. FMD predicts CV risk [10,11]. The endothelium modulates arterial stiffness, which precedes overt atherosclerosis. Arterial stiffness is associated with traditional risk factors for atherosclerosis [12] and is an independent predictor of CV events [13] mainly in selected high-risk patients with arterial hypertension (AH), diabetes mellitus (DM), and end-stage chronic renal failure [12]. Arterial stiffness is characterized by pulse wave velocity (PWV) and parameters derived from peripheral pulse wave analysis: augmentation index (Aix), central augmented, systolic, and diastolic pressure [13]. Although the relationship between aortic stiffness and CV disorders is continuous, a PWV threshold of 10 m/s has been suggested as a confirmation of hypertension-mediated organ damage in middle-aged hypertensive patients [14]. Morphological changes of the arterial wall were shown to be the most valuable markers of subclinical atherosclerosis and predictors of CV events. The current European guidelines suggest not using biomarkers or imaging modalities including detection of functional markers for CV risk stratification in populations with low or high CV risk, the added value is still questionable in those with moderate calculated risk [11]. Organ-specific biomarkers may be useful to guide therapy in specific circumstances (screening for microalbuminuria in patients with DM or AH) [14]. Coronary artery calcium (CAC) scoring, ankle-brachial index (ABI), and atherosclerotic plaque detection by carotid artery scanning are recommended with IIbB level of evidences as risk modifiers in subjects near the decisional risk thresholds to improve risk prediction [11]. CaC is a specific marker of the magnitude of atherosclerosis, and predicts future coronary events in asymptomatic patients [15]. An ABI ≤ 0.90 is associated with a two- to threefold increased risk of total and CV death. An ABI > 1.40 (medial arterial calcification) is also associated with a higher risk of CV events and mortality [16]. Carotid intima–media thickness (IMT), plaque, and stenosis are widely used as early surrogate markers of subclinical atherosclerosis and strong predictors of future deaths and CV events. The IMT is not only a measure of early atherosclerosis, but also of smooth muscle hypertrophy. The extent of carotid IMT is an independent predictor of CV diseases, but the association is not linear [17,18]. Plaques are related to both coronary and cerebrovascular events [17].

Our main objective was to investigate the relationship between microalbuminuria and various indicators of subclinical atherosclerosis in middle-aged, nondiabetic, apparently healthy subjects, with low-to-moderate estimated SCORE risk in the East Slovak region. The secondary objective was, even with the expected marginal prevalence of some markers, to evaluate the potential associations of microalbuminuria with established risk modifiers, especially with the presence of carotid plaque. Finally, based on results, we aimed to assess the role of personalized CV risk stratification including screening for microalbuminuria in high-risk European countries. To our knowledge, there are very limited data on complex analyses of the relationship between microalbuminuria and subclinical atherosclerosis in such individuals.

## 2. Patients and Methods

We included 187 apparently healthy individuals of Caucasian origin without established CV diseases, 75 men (40.1%) and 112 women (59.9%), 35–55 years old, in this cross sectional, real-life study between February 2010 and February 2014. People were invited to the study via messages advertised through various channels (e.g., social media, primary health care pratitioners, and internists, including the contact outpatient department of the study within preventive healthcare check-ups). The inclusion criteria were as follows: (a) males or females 35–55 years of age, (b) non-DM, (c) inhabitants of the East Slovak region, and (d) obtained written informed consent. Exclusion criteria included: established CV diseases, SCORE risk ≥5%, chronic kidney disorders (CKD) including estimated glomerular filtration rate (eGFR) < 1.0 mL/s/m${}^{2}$ as well as pathological urinary findings, neoplastic, hepatic, and chronic respiratory disorders, severe obesity, alcoholism, noncompliance, pregnancy, and acute inflammatory disorders. We expected approximately 400–450 eligible individuals; 311 people contacted us, wtih 256 meeting the inclusion criteria. We excluded 46 patients based on the exclusion criteria, and another 23 patiens were excluded for an incomplete clinical history or pathological ECG. Therefore, 187 apparently healthy individuals were finally enrolled in the study. This study was conducted in accordance with the Declaration of Helsinki, and the protocol was approved by the Ethical Committee of the L. Pasteur University Hospital in Košice (approval number 2020/EK/02018), and all participants gave written informed consent.

### 2.1. Data Collection

Participant examinations were conducted in the Outpatient Department of the 4th Department of Internal Medicine at L. Pasteur University Hospital and Faculty of Medicine, PJ Šafárik University in Košice, in the morning (7:30–10:00), under basal conditions. The examination itself consisted of blood and urine collection for biochemical analysis, examination of functional and morphological markers of subclinical atherosclerosis, interviews for detailed personal and family medical history with the focus on classical risk factors for atherosclerosis, current medications, measurements of body size, waist circumference, determination of 10-year fatal and total CV risk (SCORE), and resting 12-lead electrocardiogram recording. Blood and urine samples were analyzed in the relevant subdivisions of the Department of Laboratory Medicine at the L. Pasteur University Hospital. Parameters used in our work, i.e., biochemical risk markers of subclinical atherosclerosis (microalbuminuria, fibrinogen, high-sensitive C-reactive protein (hsCRP), and lipoprotein(a) (Lp(a)) and metabolic parameters (fasting glucose, glycated hemoglobin (HbA1c), uric acid, serum total cholesterol (T-C), high-density lipoprotein cholesterol (HDL-C), low-density lipoprotein cholesterol (LDL-C), triacylglycerids (TAG), and serum creatinine) have been directly determined by standard, accredited laboratory tests; eGFR were calculated according to the Modification of Diet in Renal Disease (MDRD) formula. Microalbuminuria was measured in an early morning sample. The following values were considered pathological, based on the calculations from accredited laboratory reference values: microalbuminuria = 20–200 mg/L urine in spot urine albumin sample, Lp(a) ≥ 75 mmol/L, hsCRP ≥ 3 mg/L, fibrinogen > 3.5 g/L, creatinine > 90 umol/L, eGFR < 1.5 mL/s/m${}^{2}$, uric acid > 357/428 µmol/L (males/females). Nonmodifiable risk factors for atherosclerosis as well as AH, dyslipoproteinemia (DLP), obesity/central obesity, DM, impaired fasting glucose, and metabolic syndrome have been defined according to current recommendations [11,19]. Smoking status was characterized as currently smoking ≥1 cigarette/day. To estimate a person’s 10-year risk of CV death we used the high-risk SCORE charts (low risk <1%/moderate risk ≥1% and <5%), and the total CV event risk was calculated by multiplying fatal risk ( 3× for men and 4× for women). The targeted dietary and pharmacological management of AH and DLP was satisfactory at the time of patient enrolment into the study.

### 2.2. Functional Markers of Subclinical Atherosclerosis

*Flow mediated vasodilation* was evaluated according to international recommendations [20], as the percentage increase in brachial artery diameter from basal to a maximum value during reactive hyperaemia. The right brachial artery diameter was measured with ultrasound (Philips HD 15, 7.5 MHz probe) at baseline (the mean of 4 readings) before 5 min occlusion of the blood flow to the forearm, and at the end of 1 min after release of the occlusion (the mean of 4 readings). The ultrasound transducer was positioned over the brachial artery, at a marked location, 5 cm proximal to the elbow crest. A pneumatic cuff was placed on the lower arm and was inflated 50 mmHg above the suprasystolic value for 5 min. 2D images and Doppler signals were recorded in a longitudinal section, at 5-fold magnification, and the diameter was measured at the intima–lumen interface between near and far wall. Endothelial dysfunction was confirmed at dilatation ≤4.5%.

*Determination of arterial stiffness by measuring the brachial AIx and the aortic PWV*. Experimental conditions were arranged according to the international guidelines [21]. The examinations were performed by the invasively validated device, Arteriograph (TensioMed) [22], that uses a noninvasive oscillometric method for measurement of arterial stiffness indices and some hemodynamic parameters (aortic and peripheral blood pressure). The device is equipped with a special computer program for data analysis and an infrared adapter providing communication between the registration of pressure changes in the brachial artery and a computer. The measurement is based on pulse wave analysis. The pulse wave registration occurs during the pressure measurement/cuff occlusion of the right brachial artery; a detailed description of the measurements and the algorithm for estimating indices have been published elsewhere [23]. Parameters used in our study: aortic PWV (expressed as carotid to femoral PWV (cfPWV); pathological value: ≥10 m/s) and brachial AIx (pathological value ≥10%).

*Ankle-brachial index* was measured with a portable Doppler instrument (Microdop, 5–10 MHz manually in accordance with the latest 2017 ESC Guidelines [24]. The ABI of each leg was calculated by dividing the highest ankle systolic blood pressure (SBP) by the highest arm SBP. The SBP was measured by a Doppler probe on the posterior and anterior tibial (or dorsal pedis) arteries of each foot and on the brachial artery of each arm. For the CV risk stratification we took the lowest ABI between the two legs (pathological values: ABI ≤ 0.9 and >1.4, respectively).

### 2.3. Morphological Markers of Subclinical Atherosclerosis

*Carotid IMT(CIMT) and plaque assessment.* Carotid ultrasonography (USG) (including all investigations based on USG) was performed by one experienced sonographer with acceptable intraobserver variability of measurements. Our intraobserver variability parameters were as follows: mean absolute difference = 0.085 ± 0.069; correlation coefficient = 0.88; coefficient of linear regression = 0.83; coefficient of variation = 7.2%. IMT and plaque were defined according to the Mannheim consensus [25,26,27]. Bilateral carotid arteries were scanned using high-resolution B-mode USG (Philips HD 15) with the 7.5-MHz probe in real-time, at 5× magnification. IMT was defined as the distance from the leading edge of the lumen–intima interface to the leading edge of the media–adventitia interface, and was measured on distinct plaque-free segment of the common carotid artery (ACC) far wall, 1 cm from the flow divider, in the end-diastole, at its presumed maximum thickness. Examinations were made online, using calipers, from lateral longitudinal projection. To improve accuracy, 4 measurements were taken in each segment. For the purpose of the study we used the mean value of 4 measurements for each side. Atherosclerotic plaque was defined as an endoluminal protrusion of at least 1.5 mm or a >50% focal thickening of the IMT relative to the adjacent wall segment. Plaque presence on both transverse and longitudinal planes was recorded in the ACC, bulb, and internal and external carotid arteries. Carotid artery parameters evaluated in our work: mean value of CIMT on the right, left (CIMTdx; sin), maximum value of IMT (CIMTmax), CIMT > 0.9 mm on the right/or left (CIMTbilat > 0.9), pathological value of CIMT by age and sex on the right/or left (asCIMTbilat), i.e., in men/women up to 40 years 0.57/0.51 mm, 41–50 years 0.61/0.57 mm, and over 50 years 0.70/0.64 mm [28], and carotid plaque.

### 2.4. Statistical Analysis

Patients were divided into 2 groups based on their level of microalbuminuria (with/without microalbuminuria). Patient clinical data were analyzed in the first step by means of descriptive statistical methods (frequency tables and graphs). Continuous variables are shown in the tables in the form of arithmetic mean and standard deviation (SD), and the categorical variables as an absolute number of concrete category and its relative representation (percentage of occurrence) in the sample. Analysis of differences in the continuous clinical parameters investigated, including atherosclerotic risk profile, biochemical parameters, and markers of subclinical atherosclerosis between patients with and without microalbuminuria was carried out using a nonparametric Mann–Whitney U test due to unconfirmed normal distribution of the values of the variables. Differences in the prevalences of categorical, dichotomical variables between patient groups with and without microalbuminuria we analyzed using Pearson’s chi squared test. The crude effect of the microalbuminuria (continuous values) on dichotomized/continuous biochemical, functional, and morphological parameters of subclinical atherosclerosis were tested by binary logistic/linear regression analysis. Univariate regression models were used to determine crude effects. The statistically significant effect of microalbuminuria was confirmed only on the occurrence of carotid plaque. In addition to microalbuminuria, the effect of conventional CV risk factors on the presence of carotid plaque was evaluated by binary logistic regression analysis. Variables included in the analysis were categorical (gender, risk age, positive family history, AH, DLP, smoking, central obesity, and metabolic syndrome) and continuous (age, duration of AH, waist circumference, number of risk factors, T-C, HDL-C, LDL-C, TAG, fasting glucose, HbA1c, uric acid, creatinine, and eGFR). Similar variables were used to determine the effect of predicting variables on microalbuminuria in linear regression. Positive family history was the only significant conventional risk factor associated with microalbuminuria. In spite of the marginal prevalence of carotid plaque (a strong CV risk modifier), subsequently multivariate models including all statistically significant predictors from univariate analysis were calculated, to show mutually adjusted effects on the carotid plaque. Finally, we attempted to establish a final model of the significant predictors of carotid plaque from the multivariate model using the backward (Wald) method of removing insignificant predictors in several successive steps. The final analysis included microalbuminuria, the number of risk factors, and positive family history. A value of *p* < 0.05 was considered statistically significant. The analyses were performed using the IBM SPSS 23.0 statistical software package. The statistical power (SP) of the study was: SP = 0.121; df:185; noncentrality parameter: 0.779; critical *t*: 1.973 (setting size d = 0.3; *p* value = 0.05).

## 3. Results

In the study sample of 187 clinically healthy, nondiabetic, middle-aged individuals (mean age 45.6 ± 5 years) in the Eastern Slovak region the documented prevalence of microalbuminuria was not rare (3.8%) and comparable to the literature. The prevalence of conventional risk factors in the entire group was as follows: male sex 40.1%, risk age 21.9%, positive family history 17.8%, DLP 71%, central obesity 57.4%, AH 25.8%, smoking 20.3%, and metabolic syndrome 16.7%. The estimated 10-year fatal CV risk (SCORE) was low to moderate (0.57 ± 0.87%) for the whole study group. Demographic characteristics, clinical, and laboratory data for the samples, stratified according to the presence of microalbuminuria, are shown in Table 1. The comparison of the microalbuminuria subgroups showed no differences in average values and pathological categories of monitored parameters between individuals with microalbuminuria and those with normoalbuminuria. Biochemical, functional, and structural markers of subclinical atherosclerosis in subjects with/without microalbuminuria are listed in Table 2. Data for personalized CV risk stratification are shown in Table 3.

The most frequent risk factors were DLP and central obesity. When we compared the prevalence of subclinical markers of atherosclerosis, we also did not find any significant differences between the two subgroups. There was a relatively high prevalence of subclinical atherosclerosis among subjects. Endothelial dysfunction measured by FMD (71.4%) as well as impaired perfusion of lower limbs (42.9%) detected by ABI were markers with the highest prevalence in the microalbuminuric group. Pathological values of both PWV and Aix occured in 14.3%, increased levels of both Lp(a) and hsCRP were present in 14.3%, and hyperfibrinogenemia in 16.7% in individuals with microalbuminuria. Pathological PWV and Aix were not significantly more frequent in the non-microalbuminuric group. CIMT parameters and plaque occurrence were comparable in both groups, but evident was a difference between them. Mean values of CIMT max, CIMT dx, and CIMT sin were practically the same in both groups and were in physiological ranges in group with vs. without microalbuminuria (0.62 ± 0.13 vs. 0.67 ± 0.13 mm, 0.5 ± 0.09 vs. 0.55 ± 0.1 mm, 0.55 ± 0.14 vs. 0.55 ± 0.1 mm, respectively). CIMTbilat > 0.9 mm was absent in the microalbuminuric group and was rare in the non-microalbuminuric group (1.1%), whereas asCIMTbilat was more prevalent in normoalbuminuric individuals (51.9% vs. 28.6%). Carotid plaques were more frequent in the microalbuminuric group (14.3% vs. 4.4%). Figure 1 shows the mean values ±SD and 95% confidence intervals (CI) of microalbuminuria values in individuals with and without carotid plaque (data are not shown in results).

When analyzing the effect of microalbuminuria on subclinical atherosclerosis, we have not found a statistically significant relationship with most factors. However, the higher value of microalbuminuria was significantly associated with the occurrence of carotid plaque (*p* = 0.035; OR = 1.035; 95% CI = 1.002–1.07) (Table 4). Apart from microalbuminuria, univariate logistic regression analysis revealed that the number of risk factors (*p* = 0.012), AH (*p* = 0.5), metabolic syndrome (*p* = 0.005), waist circumference (*p* = 0.036), HDL-C (*p* = 0.02), TAG (*p* = 0.003), and uric acid (*p* = 0.005) were also significantly associated with the presence of carotid plaque (variables with statistically significant effects are listed in Table 5). Among classical CV risk factors including functional renal parameters, only family history was significantly (*p* = 0.045; B = 4.68; CI 95% = 0.11–9.25) associated with microalbuminuria, with indirect support of microalbuminuria (Table 6). Although there were some statistically significant differences between men and women in some factors, gender as a variable was eliminated in a backward regression model as not an important factor in relation to changes in microalbuminuria or atherosclerotic plaque. We presented only statistically significant predictors resulting from complex models (Table 5 and Table 6). In the next step we focused on the prediction capacity of microalbuminuria together in a common model with all the statistically significant variables on the occurrence of carotid plaque (simply expressed as the number of risk factors) and the only significant determinant of microalbuminuria. From a combined simplified regression model, where variables were microalbuminuria, number of risk factors, and family history, statistical significance showed in only 2 variables: microalbuminuria and the number of risk factors (Table 7). Figure 2 illustrates the strength of correlation between microalbuminuria and non-microalbuminuria variables with the presence of carotid plaque in univariate and multivariate logistic regression.

## 4. Discussion

Our cross sectional, real-life small study in clinically healthy, middle-aged, nondiabetic individuals with/without AH, but free of confirmed CV disorders showed not rare prevalence of microalbuminuria and relatively high prevalence of subclinical atherosclerosis, presented either by biochemical, functional, or morphological markers. The higher level of microalbuminuria was associated with the occurrence of carotid plaque. No other association of microalbuminuria with subclinical atherosclerosis markers was documented. Microalbuminuria and the number of classical CV risk factors were the only factors that significantly predicted carotid plaque. To our knowledge, our study is one of the few on complex analysis of the association between microalbuminuria and various indicators of subclinical atherosclerosis in apparently healthy, middle-aged, nondiabetic individuals.

The prevalence of microalbuminuria in 187 clinically healthy individuals with low to moderate CV risk, living in the region of Eastern Slovakia, was 3.8% and comparable to the literature. DLP either treated or newly diagnosed (71%) and central obesity (57.4% ) were the most frequent risk factors. The pharmacological treatment of our study group was satisfactory. Lipid lowering medication had no significant effect on the presence of carotid plaque. In the microalbuminuric group 14.3% vs. 6.1% of subjects in the normoalbuminuric group received hypolipidemic therapy (Table 5 shows only significant parameters in association with carotid plaque). Subjects with microalbuminuria received no antihypertensive medication due to the newly diagnosed and borderline character of AH. The proportion of antihypertensive treatment (also with antisclerotic, and urinary albumin excretion reducing effect) in the non-microalbuminuric group was as follows: in 11.7% ACEI/ARB, in 7.8% betablockers, in 1.1% diuretics, and in 5.5% of patients calcium channel blockers were indicated either in monotherapy or in combination.

There was no significant difference regarding the risk profile between the microalbuminuric group and those with normoalbuminuria. DLP remained a leading risk factor in both groups. A relatively high prevalence of subclinical atherosclerosis was observed in the whole study group, presented mainly by functional markers. Biochemical markers slightly predominated in microalbuminuric subjects as well as endothelial dysfunction measured by FMD and impaired perfusion of lower limbs detected by ABI. Arterial stiffness was not significantly more frequent in normoalbuminuric subjects. Carotid artery wall characteristics were comparable in both groups, but while IMT was less affected, carotid plaque was more frequent in group with microalbuminuria. In univariate logistic regression analysis microalbuminuria, the number of risk factors, AH, metabolic syndrome, waist circumference, HDL-C, TAG, and uric acid were significantly associated with the presence of carotid plaque. Moreover, family history was significantly associated with microalbuminuria, supporting its secondary impact (via microalbuminuria) on the development of atherosclerotic lesions. In a multivariate regression model, statistically significant association with the carotid plaque showed only in microalbuminuria and the number of risk factors.

The definition of microalbuminuria varies according to the used techniques of measuring urinary albumin [5,29]. Microalbuminuria is still used in clinical practice; however, KDIGO guidelines recommend using the term “moderately increased albuminuria” instead of microalbuminuria [29,30]. In the majority of studies microalbuminuria is defined as the urinary albumin-to-creatine ratio (ACR) [31,32,33,34,35]; in other studies, it is the urinary albumin excretion (UAE) expressed as mg/24 h [36], or, as we define it, as mg/L urine, as in the PREVEND study [37]. Urinary albumin concentration (UAC) in 24 h samples is a reliable measurement in large population studies for estimating the prevalence of microalbuminuria in both sexes and irrespective of age when compared with the gold standard UAE. ACR needs both sex- and age-specific discriminator values. The diagnostic performance of measuring UAC in a spot morning urine sample in predicting microalbuminuria in subsequent 24-h urine collections is satisfactory, and comparable to that of measuring ACR [37,38,39]. We used UAC with a relevant estimation of gender (no overestimation of the prevalence of females). The prevalence of microalbuminuria is low among young people, rising with age, with a prevalence of 8–10% in the general elderly population, 20% among hypertensives, and 30% among individuals with type 2 DM (T2DM) [4,40,41]. Among the healthy (free of DM and AH, without CV disorders) 53.3 ± 9.4 year old population the prevalence of microalbuminuria was 5.4% [31]. The prevalence of microalbuminuria in an “apparently healthy” population is also similar in our study.

Verhave et al. reported [38] a higher prevalence of microalbuminuria in men than in women (male/female ratio 1.8) and underlined that the prevalence increases with age, which is more pronounced in men than in women. In the PREVEND study, the proportion of males with microalbuminuria was 53.8% [37]; we had 71.4% females and 28.6% males with microalbuminuria in a much smaller study group and there was no statistical difference between genders in the normoalbuminuric vs. microalbuminuric groups. Hillege et al. [37] documented that microalbuminuria (UAC-based) was, in the general population, associated with age and sex (male) and various CV risk factors in uni- and multivariate models. Similar associations were confirmed also for nondiabetic and nonhypertensive subjects (but without the exclusion of subjects with established CV diseases). Lian et al. [42] found the positive and highly significant correlation of microalbuminuria with HbA1c in an unadjusted model, but also in an adjusted model for age, sex, and various CV confounders; moreover, in subgroup analyses including gender, obesity, hypertension, and smoking habits, the relationship remained significant and stable. Similarly Ren et al. [43] documented, that metabolic syndrome was independently associated with greater incidences of low-grade albuminuria and CKD on unadjusted logistic regression analyses, but also after adjustment for age, sex, and potential risk factors. We did not find a significant correlation between microalbuminuria or plaque occurrence and gender.

The risk profile analysis in different studies is inconsistent according to incomplete clinical data. In comparison with other studies, our study group was smaller and included younger individuals and hypertensives, but subjects with CV disorders and DM were strictly excluded [31,32,33,34,37]. The higher prevalence of DLP in our study is related partially to a different definition of DLP in older studies [31,37].

*Functional markers of subclinical atherosclerosis and microalbuminuria*. Subclinical atherosclerotic status can reflect vascular injuries/endothelial dysfunction, ref. [35] which is associated with increased CV risk [36]. Several methods are used to assess vascular endothelial function, including FMD [36,44,45]. There are limited data about the association between microalbuminuria and FMD. Yun et al. [36] showed that FMD was significantly decreased in hypertensive patients with carotid plaque, especially in those also with microalbuminuria. Other studies documented that FMD was impaired in individuals with microalbuminuria vs. without, regardless of whether they had diabetes [46], supporting the development of endothelial dysfunction in microalbuminuria independently of DM. Indeed, several studies have shown that endothelial dysfunction precedes and predicts the onset of microalbuminuria in individuals without/with DM [47]. In our study with a similar proportion of hypertensives, nondiabetic subjects with microalbuminuria had more frequently impaired FMD and carotid plaque occurrence, in comparison with those without microalbuminuria.

Aortic PWV is an independent predictor of CV morbidity in high-risk patients, and in the general population [48,49,50]. Albuminuria was identified as an independent risk factor for PWV [35,51,52]. Increased arterial stiffness is a major cause of a wide pulse pressure and systolic hypertension, and may contribute to glomerular damage and albuminuria. Although hypertension has been significantly associated with microalbuminuria, Kohara et al. reported [53], that microalbuminuria was independently associated with PWV, indicating that the association was not merely a reflection of the increased pulse pressure and SBP. We did not find an association between microalbuminuria and parameters of arterial stiffness, probably due to the small study sample, low proportion of patients with microalbuminuria, AH, and having no diabetics in our study. However, non-microalbuminuric subjects had more prevalently abnormal PWV and Aix.

Several studies have shown an association between ACR and peripheral artery disease (PAD), predominantly in the diabetic population [35,51,54]. We confirmed a higher prevalence of PAD in the microalbuminuric group, but no relationship was found between microalbuminuria and ABI, probably due to the fact that the diabetic population per se is more sensitive to the development of vascular damage.

*Morphological markers of subclinical atherosclerosis and microalbuminuria*. Numerous community-based studies confirmed significant correlations between IMT and cardio- and cerebrovascular events [17,18,55], whatever the method and the site of IMT measurement [56], but the risk is nonlinear [17] and is more predictive in women [11]. IMT is associated with several CV risk factors as well as with the extent of atherosclerosis and end-organ damage of high-risk patients [57]. Carotid IMT > 0.9 mm is considered abnormal, but the upper limit of normality varies with age [14]. In the IRAS study an interrelationship was found between carotid IMT and microalbuminuria in nondiabetic and noninsulin dependent DM subjects [58]. In a study by Choi et al. [51] albuminuria was associated with PAD, but not with carotid plaque or carotid IMT in T2DM. Ishimura et al. [52] and Ito et al. [59] reported no association between albuminuria and carotid IMT. We found significant correlation between microalbuminuria and carotid plaque, but not with carotid IMT in nondiabetics, probably due to artery-related differences in atherosclerosis expression and involvement of different pathologic processes in IMT thickening and plaque formation.

The increased risk associated with the ACR increase is a continuum [29,60]. Low-grade albuminuria is also significantly associated with subclinical atherosclerosis [61,62]. An association betwen high-normal albuminuria and thickened IMT as well as a higher incidence of carotid plaque was documented in the large Japanese study with nondiabetic and nonhypertensive men [33]; it was also seen in Chinese community-based patients with T2 DM [54] and in the general population [63]. Liu et al. [35], in their complex, large study with participants aged 58.2 ± 9.6 years, free from CV diseases, showed that people with low-grade albuminuria (7.82 ≤ ACR < 30 mg/g) and albuminuria (ACR ≥ 30 mg/g) had higher levels of subclinical atherosclerosis (carotid: IMT > 0.6 mm or the presence of carotid stenosis as ≥50%, elevated baPWV, but not ABI), but also higher all-cause deaths independent of glycaemic status and renal function. The association of higher ACR with all-cause deaths was stronger among individuals who had concomitant subclinical atherosclerosis, DM, or more CV risk factors. They found significant positive associations of low-grade albuminuria and carotid atherosclerosis. Our results are consistent with most, but not all studies, that have demonstrated associations of ACR with indicators of subclinical atherosclerosis [62,64], but the comparison is extremely difficult due to different study design. Our study population was smaller, younger, and nondiabetic, with the exclusion of individuals with macroalbuminuria. In line with Liu et al. [35] our study also documented a higher prevalence of carotid atherosclerosis and pathological ABI but not arterial stiffness in the microalbuminuric group, which may be explained by the higher measured mean clinical blood pressure in their patients. Our study revealed the significant association between microalbuminuria levels and subclinical atherosclerosis, represented only by the occurrence of carotid plaque; another relationship, except for the borderline association with the fibrinogen level, was not found. Microalbuminuria and the number of CV risk factors were the only predictors of the occurrence of carotid plaque in our study. Interestingly, their longitudinal study showed a similar trend: the stronger risk of all-cause deaths among individuals with higher ACR and more CV risk factors.

Several studies have found an association between microalbuminuria and the presence of CaC [65,66,67]. In T2 diabetics, aged 61.2 ± 9.2 years, with relatively preserved renal function, ACR was strongly and positively associated with CAC (*p* = 0.004) and carotid artery calcium (*p* = 0.0004) [68]. In the GEA study [31], the independent risk of subclinical atherosclerosis was three times as high for healthy subjects with microalbuminuria. We did not assess CAC, but confirmed the association between carotid manifestation of atherosclerosis and microalbuminuria (not macroalbuminuria) also in nondiabetic, healthy, middle-aged subjects. In our small study, microalbuminuria (not macroalbuminuria) predicts the presence of carotid plaque in a nondiabetic, clinically healthy, middle-aged population.

The predictive power of urinary albumin levels (even within the normal range) for CV risk is independent of other CV risk factors and not only is present in high-risk individuals but also in the general population and in healthy individuals [5,29,30,69,70]. The possible link between microalbuminuria and CV risk is very complex, and reflects not only common risk factors, including insulin resistance [71,72], but rather common pathophysiologic process, such as endothelial dysfunction or chronic, low-grade inflammation [47,73]. The Steno hypothesis explains the albumin leakage as a result of widespread vascular damage and endothelial dysfunction [74]. Genetic studies also underline that the increased filtration of albumin in the glomerulus, potentially as a result of endothelial damage, and not albuminuria per se may link albuminuria to AH and increased CV risk [75]. Differences in UAE already at a very early age reflect an individual state of (patho)physiologic vascular dysfunction (determined by genetic and environmental factors) that make an individual susceptible to organ damage. Thus, albuminuria may be a useful marker in both primary and secondary prevention [5].

Our study has some limitations. The relatively low statistical power of the study is the major limitation. A real-life character of the study, relatively low prevalence of microalbuminuria and carotid plaque [76] in an apparently healthy, middle-aged population, as well as the time-consuming protocol of the study were the main limitations to increasing the statistical power of the study. Due to low statistical power, our results should be interpreted with caution as some of our results may be negative by chance. Replicating our results in a study with greater statistical power might show statistically significant associations between variables that are not statistically significant in the present study [77]. Despite this limitation, in addition to the scientific, we also highlight the clinical view with an effort to implement our findings into practice. Other limitations are: a cross sectional analysis could not show direct causal links; a relatively small number of participants, especially after selection on individuals with/without microalbuminuria; difficult comparison of our results with other studies for inconsistent use of indicators and their parameters in the evaluation of subclinical atherosclerosis; microalbuminuria was calculated in a spot morning urine sample instead of 24 h urine sample. The diagnostic performance of measuring UAC in a spot morning urine sample in predicting microalbuminuria in subsequent 24 h urine collections is satisfactory, and, moreover, comparable to that of measuring ACR (the sensitivity/specificity of UAC and ACR in spot urine predicting microalbuminuria is 69.3%/95.8% and 65.4%/97.7%, respectively). The use of spot urine collection was recommended and widely applied in population surveys [39].

## 5. Conclusions

Our results may indicate an association of microalbuminuria with the presence of carotid plaque; in addition, microalbuminuria and the number of risk factors appear to be possible predictors of the carotid plaque occurrence in nondiabetic, middle-aged, apparently healthy individuals with or without AH, suggesting that they may confer a higher risk of a future CV event. Monitoring microalbuminuria may improve the personalized cardiovascular risk assessment. Prospective studies of a larger sample should be conducted to verify the relationship between low-grade albuminuria/microalbuminuria and the presence of carotid atherosclerotic lesions.

## Figures and Tables

**Figure 1 diagnostics-11-01716-f001:**
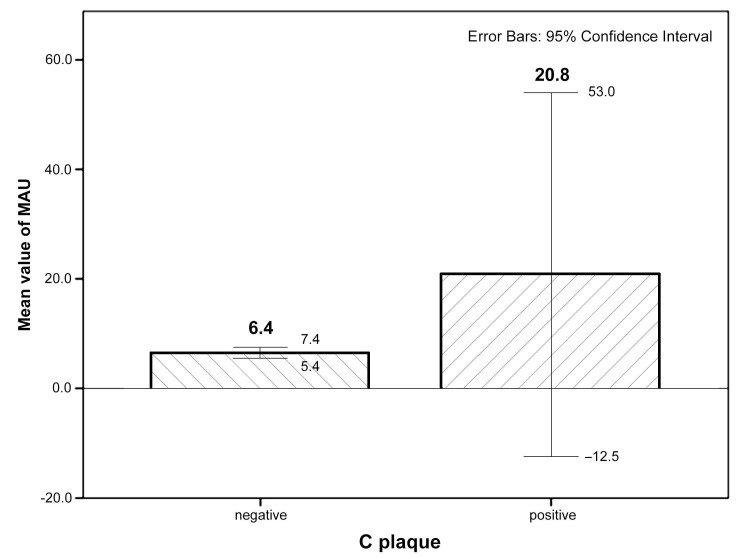
Mean values of microalbuminuria in individuals with and without carotid plaque. Note: data are shown as mean values of microalbuminuria (in mg/L) ± standard deviation; 95% confidence interval for individuals with absence (negative: 6.44 ± 6.69 mg/L; 95% CI 5.44–7.44) or presence of carotid plaque (positive: 20.77 ± 42.23 mg/L; 95% CI −12.46–53.00. C plaque: carotid plaque.)

**Figure 2 diagnostics-11-01716-f002:**
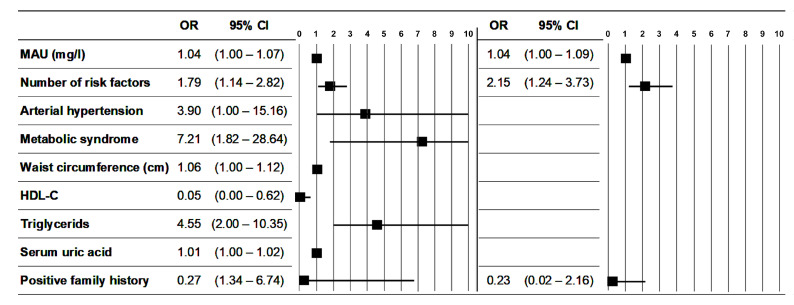
The results of univariate and multivariate logistic regression of the carotid artery plaque occurrence. After adjusting for potential confounders, microalbuminuria and the number of risk factors were identified as independent predictors for the occurrence of carotid plaque. Odds ratio and 95% confidence interval. Note: OR: odds ratio; N: number; left side: univariate logistic regression; right side: multivariate logistic regression; MAU: microalbuminuria.

**Table 1 diagnostics-11-01716-t001:** Comparison of mean values and standard deviations (SD) of continuous demographic, clinical, and laboratory parameters between groups with and without microalbuminuria assessed with *t*-test. Number of males and females between groups with and without microalbuminuria compared using Person’s chi squared test. No statistically significant differences were found.

Parameters	MAU	Non-MAU
	N = 7 Mean (SD)	N = 180 Mean (SD)
Age (yr)	44.1 ( 6.5)	45.7 ( 5.0)
Female (N/%)	5/71.4	107/59.4
Male (N/%)	2/28.6	73/40.6
BMI (kg/m${}^{2}$)	23.51 (3.15)	25.39 (4.06)
Waist circumference (cm)	83.71 (12.45)	87.78 (13.10)
SBP (mmHg)	123 (12)	124 (13)
DBP (mmHg)	80 (8)	79 (10)
Total cholesterol (mmol/L)	5.78 (1.70)	5.47 (0.96)
LDL-C (mmol/L)	3.3 (1.15)	3.25 (0.81)
HDL-C (mmol/L)	1.57 ( 0.41)	1.50 (0.36)
Triglycerides (mmol/L)	1.35 (0.88)	1.28 (0.85)
Glucose (mmol/L)	5.17 (0.61)	5.02 (0.54)
HbA1c (IFCC) (%)	3.6 (0.39)	3.4 (0.38)
Serum uric acid (µmol/L)	275.86 (79.42)	298.09 (84.56)
Creatinine (µmol/L)	85.43 (13.55)	86.63 (11.27)
eGFR (mL/s/m${}^{2}$)	1.17 (0.31)	1.15 (0.15)

Note: MAU: with microalbuminuria; non-MAU: without microalbuminuria; BMI: body mass index; SBP: systolic blood pressure; DBP: diastolic blood pressure; LDL-C: low-density lipoprotein cholesterol; HDL-C: high-density lipoprotein cholesterol; HbA1c: glycated hemoglobin; eGFR: estimated glomerular filtration rate; yr: years; N: number; SD: standard deviation.

**Table 2 diagnostics-11-01716-t002:** Comparison of mean values and standard deviations of continuous biochemical, functional, and morphological markers of subclinical atherosclerosis between groups with and without microalbuminuria assessed with t-test. No statistically significant differences were found.

Parameters	MAU	Non-MAU
	N = 7 Mean (SD)	N = 180 Mean (SD)
Microalbuminuria (mg/L) *	83.50 (88.89)	5.53 (2.53)
Lipoprotein(a) (nmol/L)	39.68 (38.47)	33.38 (36.94)
Fibrinogen (g/L)	3.14 (0.56)	2.83 ± 0.59
hsCRP (mg/L)	1.70 (1.99)	2.09 ± 2.49
FMD (%)	2.36 (9.07)	5.38 ± 9.30
RABI	1.0 (0.16)	0.97 (0.18)
LABI	1.0 (0.17)	0.94 (0.23)
Aix (%)	−22.6 (28.06)	−6.88 (28.5)
PWV (cm/s)	8.10 (1.63)	9.13 (2.49)
CIMT max (mm)	0.62 (0.13)	0.67 (0.11)
CIMT dx (mm)	0.5 (0.09)	0.55 (0.10)
CIMT sin (mm)	0.55 (0.14)	0.55 (0.10)

Note: hsCRP: high-sensitive C-reactive protein; FMD: flow mediated vasodilation; PWV: pulse wave velocity; Aix: augmentation index; RABI/LABI: ankle-brachial index right/left; CIMTmax/dx/sin: carotid artery intima–media thickness maximum value/mean value of right/left carotid artery; * microalbuminuria (selection criterium) representes *p* < 0.001.

**Table 3 diagnostics-11-01716-t003:** Comparison of prevalence and mean values of variables related to cardiovascular risk profile between groups with and without microalbuminuria. No statistically significant differences were found.

Parameters	MAU	Non-MAU
	N = 7 N/%	N = 180 N/%
Risk age	1/14.3	40/22.6
Gender (male)	2/28.6	73/40.6
Positive family history	2/28.6	31/17.4
Dyslipidemia	4/57.1	128/71.5
Central obesity	4/57.1	101/57,4
Hypertension	2/28.6	46/25.7
Smoking	3/42.9	35/19.4
Metabolic syndrome	1/14.3	30/16.9
Duration of hypertension (yr); Mean (SD)	0.14 (0.38)	0.94 (3.07)
SCORE fatal (%); Mean (SD)	0.43(0.79)	0.58 (0.87)
SCORE total (%); Mean (SD)	1.43(2.51)	1.83 (2.72)
Number of risk factors; Mean (SD)	2.57 (1.27)	2.53 (1.58)
Creatinine > 90 µmol/L	1/33.3	35/34.7
eGFR (<1.5 mL/s/m${}^{2}$)	4/57.1	13/13.1
Lipoprotein(a) (pathological)	1/14.3	23/13.3
Fibrinogen (pathological)	1/16.7	23/12.8
hsCRP (pathological)	1/14.3	39/21.8
PWV (pathological)	1/14.3	45/26.3
Aix (pathological)	1/14.3	55/32.2
FMD (pathological)	5/71.4	87/48.6
RABI or LABI (pathological)	1/42.9	27/15.2
CIMTbilat > 0.9 mm (present)	0/0	2/1.1
asCIMTbilat (pathological)	2/28.6	97/51.9
Carotid plaque (present)	1/14.3	8/4.4

Note: CIMTbilat: carotid artery intima–media thickness bilaterally; asCIMT: CIMT by age and sex on the right or left.

**Table 4 diagnostics-11-01716-t004:** Univariate (crude) effect of microalbuminuria level as predictor on markers of subclinical atherosclerosis (dependent variables), linear (B), and logistic (OR) regression coefficient and 95% confidence interval presented. Except for carotid plaque no statistically significant differences were found.

Dependent Variables		Confidence Interval 95%	
	B	Lower Bound	Upper Bound
hsCRP (mg/L)	−0.01	−0.03	0.02
Fibrinogen (g/L) *	0.01	−0.0005	0.01
Lipoprotein (a) (nmol/L)	0.12	−0.29	0.52
FMD (%)	−0.04	−0.125	0.046
PWV (cm/s)	−0.01	−0.040	0.02
Aix (%)	−0.09	−0.453	0.27
RABI	0.0004	−0.001	0.002
LABI	0.0001	−0.002	0.002
CIMT max (mm)	−0.0001	−0.002	0.001
CIMT dx (mm)	−0.001	−0.002	0.0004
CIMT sin (mm)	−0.0002	−0.001	0.001
		**Confidence Interval 95%**	
	**OR**	**Lower Bound**	**Upper Bound**
Carotid plaque (present) **	1.035	1.002	1.070

Note: B: linear regression coefficient B; * Fibrinogen: *p* = 0.068; ** Carotid plaque: *p* = 0.035.

**Table 5 diagnostics-11-01716-t005:** Variables with a statistically significant effect on the presence of carotid plaque (N = 9) in univariate logistic regression analysis. Gender is not a significant predictor.

Variables		Confidence Interval 95%		*P*
	Odds Ratio	Lower Bound	Upper Bound	
Microalbuminuria (mg/L)	1.035	1.002	1.070	0.035
Number of risk factors	1.790	1.138	2.815	0.012
Arterial hypertension (present)	3.895	1.001	15.161	0.050
Metabolic syndrome (present)	7.212	1.816	28.640	0.005
Waist (cm)	1.058	1.004	1.115	0.036
HDL-C (mmol/L)	0.046	0.003	0.620	0.020
Triglycerides (mmol/L)	4.546	1.997	10.346	0.0003
Uric acid (µmol/L)	1.013	1.004	1.023	0.005

**Table 6 diagnostics-11-01716-t006:** Results of univariate linear regression analysis with testing possible effect on microalbuminuria level by individual variables. Linear regression coefficient B and 95% confidence interval. Except for positive family history no statistically significant differences were found. Gender is not a significant predictor.

Variables		Confidence Interval 95%	
	B	Lower Bound	Upper Bound
Age (yr)	0.05	−0.28	0.39
Duration of hypertension (yr)	0.01	−0.73	0.76
Waist circumference (cm)	0.004	−0.12	0.13
Number of risk factors	0.38	−0.71	1.48
Total cholesterol (mmol/L)	0.98	−0.90	2.87
LDL-C (mmol/L)	0.42	−1.81	2.65
HDL-C (mmol/L)	−0.39	−5.28	4.49
Triglycerides (mmol/L)	1.03	−1.31	3.37
Glucose (mmol/L)	2.56	−1.11	6.24
HbA1c (IFCC) (%)	0.15	−4.67	4.98
Serum uric acid (µmol/L)	−0.006	−0.03	0.01
Creatinine (µmol/L)	−0.06	−0.21	0.090
eGFR (mL/s/m${}^{2}$)	1.91	−10.63	14.44
Gender/male (N)	−1.99	−5.46	1.48
Risk age (N)	−1.80	−5.94	2.34
Positive family history (N) *	4.68	0.11	9.25
Arterial hypertension (N)	3.22	−0.69	7.12
Dyslipidemia (N)	−0.35	−4.11	3.41
Smoking (N)	2.42	−1.79	6.64
Central obesity (N)	1.01	−2.48	4.50
Metabolic syndrome (N)	4.38	−0.14	8.89

Note: * Positive family history: *p* = 0.045; N: number.

**Table 7 diagnostics-11-01716-t007:** Final multivariate binary logistic regression model from statistically significant predictors of carotid plaque (N = 9). Odds ratio (OR) and 95% confidence interval. Significant variables were entered into a multivariate model which was then run through a backwards eliminating procedure where only mutually adjusted and still significant variables remained.

Variables		Confidence Interval 95%		*P*
	OR	Lower Bound	Upper Bound	
Microalbuminuria (mg/L)	1.04	1.003	1.09	0.034
Number of risk factors (N)	2.15	1.24	3.73	0.006
Positive family history (N)	0.23	0.02	2.16	NS

Note: NS: not significant.

## Data Availability

The datasets generated during and/or analyzed during the current study are available from the corresponding author on reasonable request. E-mail:eva.szaboova@upjs.sk.

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
