# Peer review of "Prevalence of Microalbuminuria and Its Association with Subclinical Carotid Atherosclerosis in Middle Aged, Nondiabetic, Low to Moderate Cardiovascular Risk Individuals with or without Hypertension"

_diagnostics, 2021, doi:10.3390/diagnostics11091716_

Round 1

Reviewer 1 Report

The objective of the study was to investigate the relationship between microalbuminuria (MAU) and various indicators of subclinical atheroscelrosis in middle aged, non-diabetic apparently healthy subjects, with low-to moderate estimated SCORE risk, and underline the role of microalbuminuria for CV risk stratification.

The authors conclude that MAU and the number of risk factors are significant and independent predictors of C plaque in non-diabetic, middle aged, apparently healthy individuals with or without AH, suggesting that they may confer a higher risk of future CV event.

The study population consists of 187 participants.

I have major concerns regarding the power of the study. The objective of the study was to investigate participants with microalbuminuria (MAU), but only 7 participants had MAU. The authors compare >40 different parameters between MAU (N=7) og non-MAU (N=180) and find no statistical significant differences (table 1-3).

Most of the remaining results (table 5+7) concerns predictors of carotid plaque. However this was not specified as the objective of the study – and these analyses are also hampered by only 9 participants having a carotid plaque. I can speculate if carotid plaque was chosen as a focus

based on the univariate analysis in table 4 where carotid plaque was the only significant result. If so, then I don´t find this to be a scientifically sound approach.

The conclusion states that “Our results underline the importance of monitoring albuminuria for the prevention of CV morbidity and mortality in the population.” However, with an OR of only 1.04 I don’t think this notion is supported by the data.

Other comments:

  1. The abstract has no conclusion

  1. A number of non-standard abbreviations makes the manuscript difficult to read (e.g. MAU, ATS, C plaque)

  1. The inclusion process should be described in more details. From the title low-to moderate CV risk was included but how was this done? It is stated in the method section that participants were randomly selected from the population of the East Slovak Region – I take it that the participants have volunteered for the study - how can this be random?

  1. Page 3: Carotid IMT has “acceptable intraobserver variability” – please supply data to support this notion.

  1. Statistical analysis: No power analysis has been described

  1. Table 2 first line: the statistical testing of microalbuminuria between the two groups is non-sense since this value has been used to make the two groups

  1. Table 4 last line: C-plaque is a dichotomous variable and should not be analyzed using linear regression.

  1. Table 5+7: It should be noted that the analyses are based on only 9 individuals with carotid plaque

  1. Figure 1: please add N to the figure

  1. The discussion should be shortened significantly and focus on a discussion of the findings of the study rather than a review.

Author Response

Dear reviewer

I would like to thank you for reading our article carefully and for your valuable comments and remarks. Based on feedback, we have modified and improved our contribution. We tried to apply all comments, you can find a list of corrected modifications in the attached file. 

Yours sincerely, 
Eva Szabóová

Reviewer 2 Report

Summary:

In this manuscript article the authors aimed to assess the prevalence of MAU and its relationship with subclinical atherosclerosis (ATS) in middle aged, non-diabetic, apparently healthy individuals (N=187; 40.1% men; 35-55 years old). Clinical and laboratory parameters, as well as circulating, functional (flow mediated vasodilation, ankle-brachial index, augmentation index, pulse wave velocity) and morphological markers (mean carotid intima-media thickness (CIMT), carotid plaque (C plaque) of subclinical ATS were analysed in MAU vs non-MAU group. MAU was present in 3.8% of individuals. Functional markers predominated in both groups. CIMT (mean±SD) in both groups was in range: 0.5-0.55±0.09-0.14mm. C plaque was more frequent in MAU (14.3%) vs non-MAU group (4.4%). MAU had no statistically significant effect on most markers of subclinical ATS, but the increasing value of MAU was significantly associated with the occurrence of C plaque (p=0.035; OR=1.035; 95% CI=1.002-1.07). Multiple logistic regression analysis, where variables belonged to: MAU, number of risk factors and family history, finally showed only 2 variables: MAU (p=0.034; OR=1.04; 95%CI=1.003-1.09) and the number of risk factors (p=0.006; OR=2.15; 95% CI=1.24-3.73) with independent and significant impact on the occurrence of C plaque.  Overall, this manuscript discusses a very important topic regarding the relationship between MAU and subclinical ATS in healthy population.  However, the manuscript would benefit from additional analyses and significant editing.

Major:

  1. Despite the cross-sectional method used in this article, the study could be more reliable if the number of MAU patients was not very low (Only seven patients; 3.8%). In addition, the study has failed to explain the difference between men vs. women in all MAU and non-MAU groups. Authors should analyze and include the comparison between gender in the result and discussion sections. 

  1. The writing style of this manuscript could significantly be improved and would require significant editing. 

Minor:

  1. In the abstract, the authors stated that “We aimed to assess the prevalence of MAU and its relationship with subclinical atherosclerosis (ATS) in middle aged, non-diabetic, apparently healthy individuals (N=187; 40.1% men; 35-55 years old).”  

I believe the 59.9% of women need to be included here unless the authors suggest that their findings are only reflective of the man participants. If so, the authors need to explain this clearly.

In the introduction, the authors wrote, “Endothelial dysfunction can be measured by flow mediated, endothelium-dependent vasodilation (FMD), which is currently considered to be the gold standard functional test, but has high measurement variability [6].” 

Do you mean flow-mediated dilation (FMD)? If so, please correct it throughout the manuscript.

  1. Tables 1,2 and 3 authors need to explain and show whether the comparisons between men and women were analyzed within the 187 participants. This is important since gender might also add another factor [Eur Heart J. 2016 Aug 1;37(29):2315-2381.]. 

  1. The authors wrote, “Other studies documented, that FMD was impaired in individuals with MAU vs. without, regardless of whether they had diabetes [37], supperoting the role of impaired endothelial NO synthesis in the association of MAU with CV disease risk, regardless of whether DM is present.”

Again, the manuscript is not an easy read and would benefit significantly from editing.

Author Response

(The authors gave the same response as above.)

Reviewer 3 Report

This paper investigated the relationship between microalbuminuria and various indicators of subclinical atherosclerosis in middle aged, non-diabetic apparently healthy subjects, with low-to moderate estimated SCORE risk, and underline the role of MAU for cardiovascular risk stratification. The paper has potentials but a major revision is needed. My suggestions for the authors are as follows:

Format changes needed

Please use L (not l) for the unit of measures for volumes, as in the International System it is Litter.Please revise the entire manuscript in this regard, including the Tables and Figure 2.

Table 1-3  - please remove last column which is not relevant (as the indications are the same for each parameter and insert at the final of the title of each table (NS) (in full not as abbreviation). For the first parameter in Table 2 (namely Microalbuminuria, mark it with , and explain under the table that   represents p<0.001. You may chose using * and ** and  for marking Fibrinogen and C plaques and explaining also both denotation under Table 4 - so, last column can be also removed). same for Table 6.

Figures 1 and 2 

  • They are blurred. Please replace them with a better quality ones.
  • Furthermore, in both figures, all numerical values must be written in English style (with point, not with comma and in black, not in grey - they will be more visible).
  • Title of a figure must be written bellow the figures (not above). Please see the Instructions for authors in this regard: https://www.mdpi.com/journal/diagnostics/instructions Accepted File Formats Authors must use the Microsoft Word template(on this MW template you will find how the figures must be entitled)

References in the text. A succession of references as you have I.e. in the first paragraph after figure 2 - [32] , [27],[28],[26], [29], must be rewritten in numerical order and as [27-29,32]. Please revise and correct the entire manuscript in this regard.

References section. Please check and provide just the first 10 authors et al. Please revise in this regard ref. 4, 8, 11, 12, 16, 17, 21, 22, 26, 30, 41, 51, 62, 66 (maybe I lost some of them, please check).

Content completions/corrections needed

2.1. Data collection. Please detail the inclusion and exclusion criteria of the patients in the study.

Discussion

  • Figures 1 and 2 presented in this section must be moved to Results section, as they are results!
  • The pharmacological therapy of your patients must be provided and discuss if this therapy is correlated with sub-clinical atherosclerosis. There is any correlation between insulin resistance and atherosclerosis in the patients with cardio-vascular risk factors? You may find helpful and please refer to https://doi.org/10.3390/diagnostics10050314.
  • Please describe also the potential use of SGLT-2 inhibitors in patients with cardiovascular risk factors without DM. Do any haematological parameters predict the risk of subclinical atherosclerosis? Please check and refer to  https://doi.org/10.3390/diagnostics11050850.
  • A table or graphic summarizing all predictors of survival atherosclerosis in non-diabetic non-hypertensive patients.  Did other studies confirm that microalbuminuria is associate with carotide plaque in non-hypertensive patients? - a summarising table would be very useful. 

Please remove 5. Limitations as a title of section 5. Limitation, and include the limitations paragraph as the last paragraph of Discussion (according to the same instructions for authors where is clearly indicated that: Discussion: Authors should discuss the results and how they can be interpreted in perspective of previous studies and of the working hypotheses. The findings and their implications should be discussed in the broadest context possible and limitations of the work highlighted......

Author Response

(The authors gave the same response as above.)

Round 2

Reviewer 1 Report

Thank you for the revised version. 

The very low power remains a big problem. I understand why you cannot do anything about it, but with a statistical power of only 0.12 all your negative results could very well be just by chance (a statistical type 2 error). 

Author Response

Dear  Reviewer,

Thank you very much for valuable and   critical    review of  our  MS.    Your major comment   is   discussed   below.

We hope that you will reconsider the suitibility of the revised MS for publication.

Many thanks,

Best regards,  E Szabóová  

Reviewer 2 Report

Summary:

In this manuscript article the authors aimed to assess the prevalence of MAU and its relationship with subclinical atherosclerosis (ATS) in middle aged, non-diabetic, apparently healthy individuals (N=187; 40.1% men; 35-55 years old). Clinical and laboratory parameters, as well as circulating, functional (flow mediated vasodilation, ankle-brachial index, augmentation index, pulse wave velocity) and morphological markers (mean carotid intima-media thickness (CIMT), carotid plaque (C plaque) of subclinical ATS were analysed in MAU vs non-MAU group. MAU was present in 3.8% of individuals. Functional markers predominated in both groups. CIMT (mean±SD) in both groups was in range: 0.5-0.55±0.09-0.14mm. C plaque was more frequent in MAU (14.3%) vs non-MAU group (4.4%). MAU had no statistically significant effect on most markers of subclinical ATS, but the increasing value of MAU was significantly associated with the occurrence of C plaque (p=0.035; OR=1.035; 95% CI=1.002-1.07). Multiple logistic regression analysis, where variables belonged to: MAU, number of risk factors and family history, finally showed only 2 variables: MAU (p=0.034; OR=1.04; 95%CI=1.003-1.09) and the number of risk factors (p=0.006; OR=2.15; 95% CI=1.24-3.73) with independent and significant impact on the occurrence of C plaque.  Overall, this manuscript discusses a very important topic regarding the relationship between MAU and subclinical ATS in healthy population.  However, the manuscript would benefit from additional analyses and significant editing.

Major:

  1. Response:

Despite the cross-sectional method used in this article, the study could be more reliable if the number of MAU patients was not very low (Only seven patients; 3.8%). In addition, the study has failed to explain the difference between men vs. women in all MAU and non-MAU groups. Authors should analyze and include the comparison between gender in the result and discussion sections. 

Authors:

Occurence of the MAU was natural and we could not achieve higher numbers of such patients. Relatively strict exlusion criteria also eliminated higher numbers of patients in this category. For example, this condition might be present among diabetic patients, but those patients were not included. We expected  approximately  400-450  eligible individuals,  really 311 contacted us,     256 patients met the inclusion criteria, we excluded 46 patients based on the exclusion criteria,  other 23 subjects  were  excluded for incomplete clinical history  or pathological  ECG. 

So finally only  187  subjects were enrolled into the study.  Moreover due to the real life character of the study   we had  other strong limitation – time factor  (full examination time: 2,5 hours/ 1 patient).The  number  of assessed parameters  was  another   limitation to have very large study   group. The gender was, of course, part of the analysis, and the research is not limited to men.

The frequences  in tables  indicate  the proportion of men, of course, to the women in the sample. This is standard in most studies. Due to the gender-specific predictive ability  for the   risk stratification   (in our study  mainly in  IMT correlations) it is justified to analyze the comparison between gender. We did not find significant effect of gender on studied variables. There was no difference  in  the prevalence of    women between  MAU vs non MAU groups. We  agree, it  was not  shown in results.  We completed  the values  in table 1.  We prepared  data (eg  % of menopausal women/% with  hormone  replacement therapy  in MAU  40%/20%  vs non MAU group  41.13%/2.8%)  for further subanalysis. But the relatively low number of patients with MAU (there were only 5 women) limited us in gender-specific subanalysis: such an analysis would reduce the statistical power of the study. To achieve a higher number of   patients with MAU, we would have to examine at least a 3-4 times larger group of patients, which was  unrealistic in this real life and  time-consuming examination protocol. Despite this limitation, in addition to the scientific, we would like  to  highlight the clinical view with an effort to implement our findings into practice, including the creation of a database of patients  with subclinical atherosclerosis. Table.1  Numbers of males/females between groups  with and without microalbuminuria compared  using Pearson’s chi squared test.

Response:

The authors lay out a list of reasons to justify the limitation of subjects participated or analyzed in their study.  Unfortunately, their argument is not strong enough (i.e. please read Hans L Hillege, 2002 and Meng Ren, 2021) and does not effect the low statistical power presented in this study, especially for men with MAU (only 2 people).

  1. Response:

Tables 1,2 and 3 authors need to explain and show whether the comparisons between men and women were analyzed within the 187 participants. This is important since gender might also add another factor [Eur Heart J. 2016 Aug 1;37(29):2315-2381.].

Authors:

We did not find significant effect of gender on studied variables.  Thats why we presented only effect of tested variables on dependent variables.  Limitations of the small MAU group including women with MAU  have been explained above.Table 1 was corrected,   Table  2: gender-specific comparison  has limitations,  Table 3: the same, and is dealing  with  cardiovascular risk profile in general (male gender).  

Response-2:

These tables are interesting, but the data and conclusion presented here are over interpreted.  With only 2 male analyzed the results need more subjects to be statistically feasible.

Author Response

Dear  Reviewer,

Thank you very much for valuable and   critical    review of  our  MS.    Your major comments are   discussed   below.

We hope that you will reconsider the suitibility of the revised MS for publication.

Many thanks,

Best regards,  E Szabóová  

Reviewer 3 Report

All the modifications have been done according to my requests. i recommend publication.

Author Response

Dear  Reviewer,

Thank you very much for valuable and   critical    review of  our  MS and  your  

approval.

Many thanks,

Best regards,  E Szabóová  

Round 3

Reviewer 2 Report

N/A